# ON THE EXPLOITATIVE BEHAVIOR OF ADVERSARIAL TRAINING AGAINST ADVERSARIAL ATTACKS

## ABSTRACT

Adversarial attacks have been developed as intentionally designed perturbations added to the inputs in order to fool deep neural network classifiers. Adversarial training has been shown to be an effective approach to improve the robustness of image classifiers against such attacks especially in the white-box setting. In this work, we demonstrate that some geometric consequences of adversarial training on the decision boundary of deep networks give an edge to certain types of black-box attacks. In particular, we introduce a highly parallelizable black-box attack against classifiers equipped with an $\ell_2$ norm similarity detector, which exploits the low mean curvature of the decision boundary. We use this black-box attack to demonstrate that adversarially-trained networks might be easier to fool in certain scenarios. Moreover, we define a metric called robustness gain to show that while adversarial training is an effective method to improve the robustness in the white-box attack setting, it may not provide such a good robustness gain against the more realistic decision-based black-box attacks.

## 1 INTRODUCTION

It is known in the literature that adversarial training can make deep neural networks more robust Madry et al. (2018); Shafahi et al. (2019); Wong et al. (2019) against adversarial attacks Goodfellow et al. (2014); Carlini & Wagner (2017); Moosavi-Dezfooli et al. (2016); Szegedy et al. (2013). Arguably, adversarial training can be assumed as one of the most effective techniques for robustness improvement Athalye et al. (2018). Moreover, it is empirically shown in Moosavi-Dezfooli et al. (2019) that adversarial training causes the boundary of the image classifiers to become flatter (less curved) compared to normally-trained ones.

Adversarial attacks can be executed in the white-box setting Carlini & Wagner (2017); Goodfellow et al. (2014); Moosavi-Dezfooli et al. (2016), score-based black-box setting Chen et al. (2017); Ilyas et al. (2018); Narodytska & Kasiviswanathan (2016) or decision-based black-box setting Brendel et al. (2018); Chen et al.; Cheng et al. (2019); Liu et al. (2019); Rahmati et al. (2020). The attacker's level of information about the classifier plays a key role on the quality of the generated adversarial examples. In order to craft an adversarial perturbation in a decision-based black-box setting, the critical information is mostly the normal vector to the decision boundary. In this setting, the estimation of the normal vector is conducted with carefully designed fine-tuned queries at a boundary point of the image classifier, usually based on the linearization of the boundary. Chen et al.; Cheng et al. (2019); Liu et al. (2019); Rahmati et al. (2020). The objective of such black-box attacks is typically to reduce the number of queries as much as possible with an efficient estimate of the normal vector. However, to make sure that this linearization approximation is valid at a boundary point, the $\ell_2$ distance of these queries with the boundary point should be small enough. As a result, such similar queries can be detected using a $k$-nearest neighbours (KNN) similarity detector as in Chen et al. (2020). Moreover, the efficient estimation of the normal to the decision boundary heavily relies on the assumption that the boundary of the image classifier has a low mean curvature in the vicinity of input samples Fawzi et al. (2016); Moosavi-Dezfooli et al. (2019). Therefore, such estimators are expected to work better if the decision boundary is less curved. Interestingly, it is empirically shown that adversarial training leads to neural networks with flatter decision boundaries, compared to the boundaries learned through regular training methods Qin et al. (2019); Moosavi-Dezfooli et al. (2019). We will show that this characteristic of the adversarially-trained networks indeed gives an edge to black-box attacks.

The goal of this paper is to show some evidence that although the adversarial training improves the robustness of deep image classifiers effectively against the minimal-norm perturbation white-box attacks, it becomes less effective in more practical attack settings. In particular, we propose a parallelizable attack against classifiers equipped with an $\ell_2$ norm similarity detector to demonstrate that adversarially-trained networks might even be fooled with smaller $\ell_2$ norm for a given query budget compared to regularly-trained networks due to their excessive linear behavior. That is, decision-based black-box attacks can exploit the excessive flatness caused by adversarial training. In addition, we define a metric called *robustness gain* as the ratio of $\ell_2$ norm of adversarial perturbation required to fool the adversarially-trained network to that required for the regular network. We observe that the level of information available to the attacker about the classifier impacts the robustness gain; in particular, the robustness gain increases when the information available to the attacker increases (e.g., from black-box to white-box, or by increasing the number of queries in the black-box setting). We summarize the contributions of this paper as follows:

- We empirically show that there is an interesting trade-off between adversarial training and the attack's effectiveness to fool the classifier. Moreover, we demonstrate that this trade-off is even more critical in certain black-box attacks which rely on the estimation of the normal vector at the boundary.

- We introduce a highly *parallelizable* attack which is effective against a classifier equipped with a query similarity detector based on $\ell_2$ norm. Using normal vectors estimated at multiple points on the boundary, we develop an attack which is, interestingly, more effective against an adversarially-trained network as compared to a regular network.

- We define a metric called *robustness gain* as the ratio of $\ell_2$ norm of adversarial perturbations required for the robust network to that for the regular network. We show that while adversarial training is an effective approach against minimum perturbation white-box attacks, it may not provide a good robustness gain against black-box attacks.

The rest of the paper is organized as follows. In Section 3, the problem setting, the similarity detector, and the multi-point normal vector estimator are introduced. In Section 4, the performance of our proposed multi-point attack and the state of the art black-box attacks is analyzed for the adversarially-trained network compared to the regular one. In Section 5, we evaluate the effectiveness of adversarial training against a minimum perturbation white-box attack and finally Section 6 concludes the paper.

## 2 RELATED WORK

**Adversarial Training** The basic idea of adversarial training is to create and then incorporate adversarial examples into the training process Szegedy et al. (2013); Goodfellow et al. (2014). In Madry et al. (2018), authors show an effective version of an adversarially-trained network to improve robustness against white-box attacks. In Shafahi et al. (2019), the authors proposed a so-called "*free*" version of adversarial training with a cost nearly as equal as natural (regular) training. Their key idea is to update both the model parameters and image perturbations using one simultaneous backward-pass. Recently, in Wong et al. (2019), the authors discovered that adversarial training can be conducted in a cheaper manner using the fast gradient sign method (FGSM) Goodfellow et al. (2014) added with random initialization. This approach can be useful to adversarially train large datasets such as ImageNet much faster.

**Adversarial attacks** Adversarial attacks can be executed in different categories depending on the attacker's level of information including white-box setting Carlini & Wagner (2017); Goodfellow et al. (2014); Moosavi-Dezfooli et al. (2016); Szegedy et al. (2013), score-based black-box setting Chen et al. (2017); Ilyas et al. (2018); Narodytska & Kasiviswanathan (2016) or decision-based black-box scenarios Brendel et al. (2018); Chen et al.; Cheng et al. (2019); Liu et al. (2019); Rahmati et al. (2020). In order to craft an adversarial example, the critical information is the normal vector to the decision boundary of the classifier. The most successful black-box attacks directly estimate the normal vector with linearization of boundary. For example, the HSJA Chen et al. deploys the gradient direction estimation information. In Liu et al. (2019); Rahmati et al. (2020), authors locally approximated the decision boundary with a hyper-plane, and searched the closest point on the hyper-plane to the benign input as the perturbation.

**Black-box defenses** The generation of an adversarial example requires the black-box attacker to submit multiple similar queries to the target model. Thus, the main idea of most of black-box defences is to detect such a similarity across multiple queries. In Chen et al. (2020), authors propose to equip an existing classifier with a detection component, which stores the similarity vectors for all incoming queries, computed by a pre-trained similarity encoder. For each new query, it computes the $k$-nearest-neighbor distance between it and all other vectors in the memory. Blacklight Li et al. (2020) computes a compact set of one-way hash values for each query image that form a probabilistic fingerprint. The variants of an image make almost identical fingerprints, which makes it robust against manipulation. In Byun et al. (2021), the authors introduce Small Noise Defense (SND) in which even a small additive input noise can neutralize most query-based attacks.

## 3 PROBLEM STATEMENT AND MULTI-POINT ATTACK

**Motivation and background** One of the most challenging settings to perform adversarial attack to image classifiers is when the attacker only has access to the top-1 label of the classifier, where the attacker's level of information from the image classifier is the least. A *query* is a request that results in the top-1 label of an image classifier for a given input. The state-of-the-art attacks try to obtain the smallest possible $\ell_p$ norm of the perturbations with efficient use of queries to the image classifier Brendel et al. (2018); Chen et al.; Cheng et al. (2019); Liu et al. (2019); Rahmati et al. (2020). An implicit assumption here is that the image classifier is *naive* enough to respond to multiple consecutive similar queries with no complain. This is a strong assumption which is in contrast with the common sense in terms of security. In practice, the defender can take advantage of such characteristic of the queries to detect the suspicious set of queries. In addition, all these attacks perform in an iterative manner which can be time consuming even for a powerful attacker with lots of processing power. Having a parallelizable attack can expedite the running time of the attack which can be critical in certain scenarios that attacker should act as quick as possible. Our goal is to propose a highly parallelizable attack to fool the classifier equipped with an $\ell_2$ similarity detector by taking advantage of the low mean curvature of the decision boundary of state-of-the-art deep classifiers.

**Similarity detector, simple yet effective defense** As in Chen et al.; Cheng et al. (2019); Liu et al. (2019); Rahmati et al. (2020), most of the state-of-the-art black-box attacks generate queries with additive Gaussian or Uniform noises to a boundary point to estimate the normal to the decision boundary. Inherently, these types of estimators need similar, i.e. very close queries in terms of $\ell_2$ distance to make sure the linearization assumption is valid. However, such similar queries can be simply detected by a $k$ nearest neighbour (KNN) similarity detector with $k \ll N$, where $N$ is the query budget for the attacker. Thus, the $k$ nearest neighbours of the given query $i$ are obtained among the queries stored in a buffer. In particular, for a given query $i$, the classifier computes the average $\ell_2$ distance between the query $i$ and its $k$ nearest neighbors $d_i$ defined as in Chen et al. (2020):

$$d_i = \frac{1}{k} \sum_{t=1}^{k} d_{i,t}, \tag{1}$$

where $d_{i,t}$ is the $\ell_2$ distance of the given query $i$ with its nearest neighbours $t$ where $1 \le t \le k$. Then, by introducing the detection threshold $\delta$, if $d_i < \delta$, an attack is detected and the user is blocked. The value for $\delta$ should not be too small so that none of the queries get captured and also not too large to detect some clean queries as a false attack. Although, the computational complexity of of the KNN detector may be high, it is quite effective and simple in detecting the queries generated by the techniques deploying normal vector estimation as in HopSkipJump Chen et al., GeoDA Rahmati et al. (2020), Sign-OPT Cheng et al. (2019), and qFool Liu et al. (2019). We stress here that the goal of the paper is not to outperform the state-of-the-art black-box attacks, but rather to provide certain practical scenarios in which the flatter boundary of the adversarially-trained networks are exploitable by the black-box attacker. Therefore, the question is:

*Is there a way to design a parallelizable attack to generate a query-efficient $\ell_2$-norm-minimized perturbation with total $N$ queries against a classifier equipped with a similarity detector of $k$ nearest neighbours with threshold $\delta$?*

We show that the answer to this question is affirmative and introduce a multi-point normal estimator which evades the similarity detector.

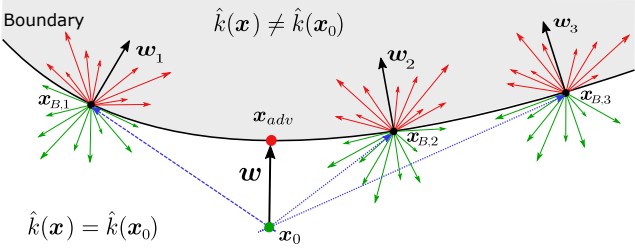

Figure 1: Multi-point normal vector estimation.

**Problem statement**  We consider a trained $L$-class classifier with parameters $\theta$ represented as $f :$ $\mathbb{R}^d \to \mathbb{R}^L$, where $\boldsymbol{x} \in \mathbb{R}^d$ is the input image and $\hat{k}(\boldsymbol{x}) = \text{argmax}_k \, f_k(\boldsymbol{x})$ is the top-1 classification label where $f_k(\boldsymbol{x})$ is the $k$-th component of $f(\boldsymbol{x})$ corresponding to the $k$-th class. The attacker's goal is to minimize the $\ell_2$ norm as much as possible with a limited budget of $N$ queries while it cannot get detected with the similarity detector. We define the optimization problem as:

$$\min_{\boldsymbol{v}} \quad \|\boldsymbol{v}\|_2 \tag{2}$$
$$\text{s.t.} \quad \hat{k}(\boldsymbol{x} + \boldsymbol{v}) \neq \hat{k}(\boldsymbol{x}),$$
$$d_i > \delta, \forall i \leq N.$$

The last constraint ensures that the queries are not detected by the $\ell_2$ similarity detector. Without the similarity detector constraint, problem equation 2 is already solved in the literature with different approaches Brendel et al. (2018); Chen et al.; Cheng et al. (2019); Liu et al. (2019); Rahmati et al. (2020). The main idea is to obtain a point on the boundary of the image classifier and estimate the normal to the decision boundary at this point. However, with the presence of the similarity detector, the normal vector estimation becomes more challenging for the attacker as it can not generate large number of queries with small $\ell_2$ norms to the neural network.

### 3.1  Multi-point normal vector estimator

We employ the fact that the decision boundaries of the state-of-the-art deep networks have a low mean curvature in the vicinity of inputs Fawzi et al. (2016; 2018). Therefore, to simplify equation 2, we locally approximate the boundary of image classifier at multiple boundary points with a hyperplane as:

$$\min_{\boldsymbol{v}} \quad \|\boldsymbol{v}\|_2 \tag{3}$$
$$\text{s.t.} \quad \boldsymbol{w}^T(\boldsymbol{x} + \boldsymbol{v}) - \boldsymbol{w}^T \boldsymbol{x}_{B,j} = 0, \; \forall j \leq M$$
$$d_i > \delta, \; \forall i \leq N,$$

where $\boldsymbol{x}_{B,j}$ is the $j$-th boundary point and $M$ is the number of boundary points obtained using a binary search. Having a single boundary point $M = 1$ and spend all $N$ queries at $\boldsymbol{x}_{B,1}$ to estimate the normal to the decision boundary can make the attack parallelizable. However, such an attack can be simply detected by the similarity detector. Also, iterative attack is not desirable since at each iteration the boundary points distances decreases. Thus, we need to design an estimator in which the queries cannot be detected by the similarity detector.

**Multi-point normal estimator**  To alleviate the aforementioned problem, we propose the *multi-point normal vector estimator* in which the queries are generated on multiple boundary points ($M \gg 1$). The key idea is to distribute $N$ queries to estimate the normal vector to the boundary using multiple boundary points, rather than spending all $N$ queries on just a single point. Apparently, a considerable portion of queries are allocated to obtain the boundary points along with binary search. This is the cost imposed by query detector to the attackers. This estimator is both parallelizable and, as we will see, successful against a classifier equipped with an $\ell_2$ similarity detector.

As seen in Fig. 1, starting from the original image $\boldsymbol{x}_0$, one can find $M$ points on the boundary denoted by $\boldsymbol{x}_{B,1}, \boldsymbol{x}_{B,2}, \ldots, \boldsymbol{x}_{B,M}$. Similar to the method proposed Chen et al.; Liu et al. (2019); Rahmati et al. (2020), the boundary points can be obtained using binary search along several random

---

**Algorithm 1:** Multi-point attack

---

1  **Inputs:** Original image $\boldsymbol{x}_0$, query budget $N$.

2  **Output:** Adversarial example $\boldsymbol{x}^{\text{adv}}$.

3  Obtain the optimal number of boundary points $M^*$ by $M^* = \frac{N-\beta}{k-b-\gamma^*}$.

4  Obtain $M^*$ starting point on the boundary $\boldsymbol{x}_{B,1}, \boldsymbol{x}_{B,2}, \ldots, \boldsymbol{x}_{B,M^*}$.

5  Estimate normal $\hat{\boldsymbol{w}}$ with equation 5.

6  Push the original image $\boldsymbol{x}_0$ towards the boundary in the direction of $\hat{\boldsymbol{w}}$.

7  $\hat{r} \leftarrow \min\{r' > 0 : \hat{k}(\boldsymbol{x}_0 + r'\hat{\boldsymbol{w}}) \neq \hat{k}(\boldsymbol{x})\}$

8  $\boldsymbol{x}^{\text{adv}} \leftarrow \boldsymbol{x}_0 + \hat{r}\hat{\boldsymbol{w}}$

---

directions, with $b$ queries on average per boundary point Normally, these $b$ queries are quite close to each other and can be assumed as *bad* queries which we have to minimize in our design as much as possible (from the attacker perspective). In general, the close queries are not desirable since they can reduce the KNN mean $d_i$ for a given query $i$ and increase the chance of getting detected by the similarity detector. On the other hand, the most informative queries are the ones with small distances with one another which creates an interesting trade-off. We assume that the attacker knows that the query detector deploys $k$ nearest neighbours to compute $d_i$ for each query $i$. Here, without loss of generality, we assume that the closest boundary point to $\boldsymbol{x}_i$ is $\boldsymbol{x}_{B,j}$. The similarity detector observes three types of queries when computing its $k$ nearest neighbours for query $i$. The first type are the queries used to obtain the boundary point with added Gaussian noise along with binary search. The second type are the ones deployed to estimate the normal to the boundary at $\boldsymbol{x}_{B,j}$. Finally, the type three distances for query $\boldsymbol{x}_i$ are obtained between the $\boldsymbol{x}_i$ and its second closest boundary point to $\boldsymbol{x}_i$. At each boundary point $\boldsymbol{x}_{B,j}, 1 \leq j \leq M$, the attacker can allocate $n = k - b - \gamma$ queries to estimate the normal to the boundary at point $\boldsymbol{x}_{B,j}$, where $b$ is the average number of type II queries, and $\gamma$ is the number of type III queries. Thus, the number of required boundary points can be obtained by:

$$M = \frac{N-\beta}{k-b-\gamma} \tag{4}$$

where $\beta$ is the number of queries required to push the $\boldsymbol{x}_0$ towards the direction of the estimated normal vector towards the boundary (step 6 of the Algorithm 1). At each boundary point $j$, the boundary is locally approximated with a hyperplane $\boldsymbol{w}_j^T(\boldsymbol{x} - \boldsymbol{x}_{B,j}) = 0$. In order to estimate the normal vector $\boldsymbol{w}_j$, the key idea is to generate $n$ samples $\boldsymbol{\zeta}_i, i \in \{1,\ldots,n\}$ from a multivariate normal distribution $\boldsymbol{\zeta}_i \sim \mathcal{N}(\boldsymbol{0}, \boldsymbol{\Sigma})$ which results in queries with the form of $\boldsymbol{x}_{B,j} + \boldsymbol{\zeta}_i, \forall i \in N$ Rahmati et al. (2020). The estimator $\hat{\boldsymbol{w}}_j$ of $\boldsymbol{w}_j$ with $n$ queries is $\hat{\boldsymbol{w}}_j = \frac{1}{n}\sum_{i=1}^{n} z_i \boldsymbol{\zeta}_i$, where $z_i = 1$ if $\hat{k}(\boldsymbol{x}_{B,j} + \boldsymbol{\zeta}_i) \neq \hat{k}(\boldsymbol{x}_{B,j})$ and otherwise $z_i = -1$. Eventually, the average normalized direction of estimated normal vector over all the boundary points can be given as:

$$\hat{\boldsymbol{w}} = \frac{\Sigma_{j=1}^{M} \hat{\boldsymbol{w}}_j}{\|\Sigma_{j=1}^{M} \hat{\boldsymbol{w}}_j\|}. \tag{5}$$

After estimating the normal to the boundary $\hat{\boldsymbol{w}}$, we push the original image $\boldsymbol{x}_0$ towards the boundary in the direction of $\hat{\boldsymbol{w}}$ with amplifying the magnitude of the vector. The final multi-point attack is summarized in Algorithm 1.

### 3.2  NUMBER OF BOUNDARY POINTS

In this section, our goal is to have an estimate for the optimal number of boundary points. The number of boundary points $M$ should not be too small resulting in close queries that will get detected by the similarity detector, and must not be too large that wastes the number of queries (as obtaining each boundary point costs $b$ queries on average). We assume that the attacker has the information about $k$ and $\delta$. Thus, by computing the KNN distance in the design step of the attack, the attacker can evade the similarity detector. As mentioned previously, for a new query $i$, the $k$ nearest neighbour queries and their distances to $\boldsymbol{x}_i$ can be categorized into three types. We aim to have an estimate for the optimal number of boundary points. For more details about mean distance of queries ($\mu_i$), refer to Appendix A.1.

Having the mean distances ($\mu_i$) of queries in hand, the attacker can have an estimate on the number of type II queries at each boundary point to get an estimation of the normal vector of the boundary

without getting detected by the similarity detector. The number of queries at each boundary point should not be too large so that the similarity detector can detect the queries due to the small $d_i$. Moreover, it should not be too small to incur excessive overhead in finding boundary points. Thus:

$$d_i = \frac{1}{k} \sum_{t=1}^{k} d_{i,t} = \frac{1}{k}(b\mu_1 + n\mu_2 + \gamma\mu_3) = \delta + \lambda, \tag{6}$$

where $\lambda$ is a margin to the threshold to make sure that the average distance can not be detected by the similarity detector. We may ignore the $n\mu_2$ term as it is typically small compared to $\mu_1$ and $\mu_3$. Having this approximation, the optimal value for $\gamma$ can be obtained as $\gamma^* = \frac{k(\delta+\lambda)-b\mu_1}{\mu_3}$: It simply can be seen that if $\mu_3$ is large, then the number of type II queries $n$ per boundary point increases which results in more efficient deployment of queries. The optimal number of boundary points can be obtained by plugging $\gamma^*$ into equation 4 as $M^* = \frac{N-\beta}{k-b-\gamma^*}$. Based on this, by increasing the number of type III distances which results in increasing the KNN mean, one can see that the number of required boundary points to satisfy $d_i = \delta + \lambda$ decreases as well.

## 4 EFFECTIVENESS OF ADVERSARIALLY-TRAINED NETWORKS AGAINST BLACK-BOX ATTACKS

In this section, our goal is to evaluate the effectiveness of adversarial training against decision-based black-box attacks in which the attacker has only access to the output label of the image classifier for a given input. We evaluate our experiments on a pre-trained ResNet-50 He et al. (2016) called *regular* network and the adversarially-trained ResNet-50 Madry et al. (2018) called *robust* network throughout this section. We consider 300 correctly classified images by both networks which are randomly selected from the ILSVRC2012's validation set Deng et al. (2009).

### 4.1 MULTI-POINT ATTACK AGAINST SIMILARITY DETECTOR

Here, we evaluate the performance of our novel multi-point attack and an iterative attack[1] (GeoDA Rahmati et al. (2020)) on an adversarially-trained network equipped with an $\ell_2$ similarity detector in Fig 2a. For the similarity detector, we assume $k = 100$, $\delta = 5$, $\zeta = 5$. We also set $n^{\text{rbst}} = 45$ and $n^{\text{reg}} = 30$. We deploy uniform GeoDA with 400 queries per iteration. It can be observed that the iterative attack can be detected after a few number of queries as similarly detector's KNN mean drops quickly below $\delta + \zeta$. However, the multi-point attack by-pass the similarity detector by keeping the KNN mean $d_i, \forall i$ greater than the detection threshold by distributing the queries over the distant boundary points.

In Fig. 2b, we compare the $\ell_2$ norm of single-point attack ($M = 1$, which is highly parallelizable, but not successful against the similarity detector) and our proposed multi-point attack on both regular and adversarially-trained networks with respect to number queries. For the single iteration attack, the performance of attack on regular network is better compared to the robust network as expected. However, for the extension to multi-point attack which can successfully evade the similarity detector, we interestingly see that the performance of the multi-point attack on the robust network is almost the same as the performance of the single point attack. However, for the regular network, the performance of the multi-point attack is much worse compared to the one for the single point attack. Thus, interestingly, *adversarially-trained network can be fooled with smaller $\ell_2$ norm with the same amount of queries*. Also, we observe that the trend of the convergence for the regular network is more noisy which is due to non-smoothness of the boundary.

The reasons behind the above observations are twofold. First, it is quite intuitive that the multi-point normal estimator can preform better on smoother boundaries. In particular, the flatter the boundary the more aligned the directions of the normal vectors at different boundary points are. In the extreme case that the boundary is a hyper-plane, the normal vectors to the boundary on all over the hyper-plane are in the same direction. Thus, this results in a better estimation of the normal vector to the boundary for the classifier in black-box setting when the network is more is adversarially-trained (boundary is flatter).

---

[1]Note that most of the state of the art attacks in which they estimate the normal to the decision boundary are iterative and follow the similar procedure. Thus, comparing with one of them is sufficient for this experiment.

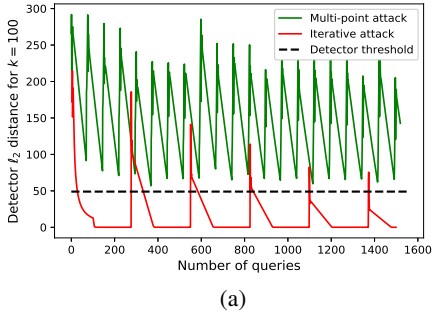 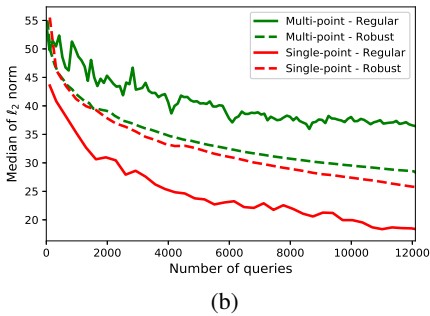

(a)            (b)

Figure 2: (a) KNN query similarity detector $\ell_2$ distance for iterative and multi-point estimator. (b) Performance of single-point and multi-point attacks on both regular and robust ResNet50.

Second, as shown in equation 9, the larger the type III mean distance $\mu_3$ is, the less number of type III queries $\gamma^*$ is needed to satisfy equation 8. In equation 4, the larger $\gamma^*$ follows with fewer number of boundary points $M^*$. This leads to a more efficient deployment of the queries in the robust network as obtaining each boundary point impose the cost of approximately $b$ queries per boundary point to the attacker. We empirically show in the Table 1 that the mean distance of the points on the boundary $\mu_3$ is larger for the adversarially-trained network compared to the one for regular network. As discussed above, this can reduce the number of required boundary points to satisfy $d_i = \delta + \lambda$ in equation 8 which benefits the attacker. Thus, we have $\mu_3^{\text{rbst}} > \mu_3^{\text{reg}}$ which leads to $\gamma^{\text{rbst}} < \gamma^{\text{reg}}$ in equation 9 and also $M^{\text{rbst}} < M^{\text{reg}}$. As a result in equation 4, we have $n^{\text{rbst}} > n^{\text{reg}}$ which results in larger number of queries at each boundary point. The smaller number of boundary points is beneficial for the attacker as it can save on average $b$ queries per boundary point. Moreover, larger number of queries at each boundary point $n^{\text{rbst}} > n^{\text{reg}}$ can increase the accuracy of the normal estimation at each boundary point.

**Black-box attacks performance evaluation**     We compare the performance of black-box attacks HSJA Chen et al., GeoDA Rahmati et al. (2020), boundary attack (BA) Brendel et al. (2018) on both regular and robust ResNet-50 networks in Fig. 3a. For a given query budget, the $\ell_2$ norm of perturbations for the attacks against the robust network is larger compared to that of the regular network as expected. However, an interesting observation is that while GeoDA has almost the same $\ell_2$ norm as HSJA for the regular network, it provides smaller $\ell_2$ norm for perturbations against the robust network compared to HSJA for a fixed amount of queries. The reason for this phenomenon is that GeoDA is *explicitly* built based on the assumption that the boundary of the classifier has a low mean curvature. On the other hand, adversarially trained-networks has flatter decision boundaries which actually gives an edge to GeoDA. Thus, to attack robust networks more efficiently, it is beneficial for the attackers to deploy attacks exploiting the flatness of the decision boundary.

**Robustness gain**     Here, our goal is to evaluate how much adversarially-trained networks can improve the robustness under various kind of attacks. We plot the *robustness gain* for different attacks in Fig. 3b. The larger the $\eta$ for a given attack is, the better the adversarial training can improve the robustness compared to the case of the regular network. In Fig. 3b, it is observed that $\eta$ is equal to around 17 (see Table 2) for the white-box attack DeepFool (DF) Moosavi-Dezfooli et al. (2016) which is a quite good improvement. Having a black-box setting, we evaluate the $\eta$ for HSJA Chen et al., GeoDA Rahmati et al. (2020), boundary attack Brendel et al. (2018), single iteration and multi-point attacks as well. First, it can be seen that, in general, the robustness gain is lower than that of the DeepFool. Second, by extracting more information from the image classifier (more queries), the robustness gain increases. For the single iteration and multi-point attack, $\eta$ is much lower compared to the that of other black box attacks with a small increasing slope along with increasing the number of queries. It may imply that the less information you know about the image classifier, i.e., the more practical the attack scenario is, the less the adversarially-trained network can improve the robustness. That being said, adversarial training for the deep image classifiers is much more effective for white-box scenarios.

**Mean distance of boundary points**     The mean distance of boundary points $\mu_3$ is a critical design parameter for the multi-point attack as it determines the number for boundary points. Here, we have done experiments to measure the $\mu_3$ for both networks with different number of boundary points $M$. We start with a boundary point $\boldsymbol{x}_{B,1}$ and obtain the $\ell_2$ norm to this point from all other boundary

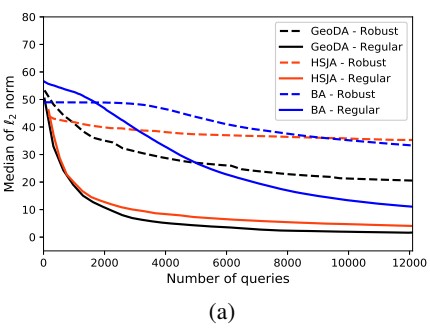 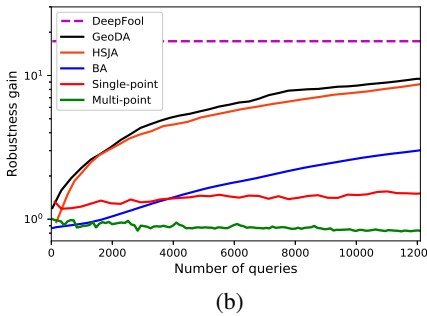

|(a)|(b)|

Figure 3: (a) Performance comparison of different black-box attacks for both regular and robust ResNet50. (b) The robustness gain for $\ell_2$ norm under different attack scenarios.

points. The average of such a distance over 300 correctly classifier images by both networks is reported in Table 1. It is observed that the $\mu_3$ is large for the robust network compared to that of regular network. Moreover, we can see that the value of $\mu_3$ even for $M = 2$ is very close to the true mean. Thus, the estimate of the $\mu_3$ can be cheaply obtained with only $M = 2$ since in the high-dimensional regime, the $\ell_2$ distances are close enough.

|  | $M = 2$ | $M = 10$ | $M = 100$ |
|---|---|---|---|
| Regular He et al. (2016) | 67.01 | 67.38 | 69.34 |
| Adv. trained Madry et al. (2018) | 90.11 | 88.47 | 89.71 |

Table 1: Mean distance of boundary points $\mu_3$ on both robust and adversarially-trained networks for different number of boundary points averaged over 200 samples.

## 5 ADVERSARIAL TRAINING AGAINST MINIMAL-NORM PERTURBATION WHITE-BOX ATTACK

We already discussed the effectiveness of the adversarially-trained network with respect to number of required queries against decision-based black-box attacks in query-limited regime. In the white-box scenario, we evaluate the effectiveness through the number of required iterations for the convergence of a minimal $\ell_2$ norm perturbation white-box attack. To this end, we choose a minimal $\ell_2$ norm white-box attack DeepFool Moosavi-Dezfooli et al. (2016) and compare its performance on an adversarially-trained Madry et al. (2018) and a regular ResNet-50 in Table 2.

**DeepFool performance** To this end, we choose a minimal $\ell_2$ norm white-box attack Deep-Fool Moosavi-Dezfooli et al. (2016) and compare its performance on an adversarially-trained Madry et al. (2018) and a regular ResNet-50 in Table 2. The main reason we choose DeepFool is its dependence on linearizing the output function of the classifier. The algorithm starts with locally linearizing the output function of the classifier and repeats such an approximation iteratively to compensate for the effect of the non-linearity of the output function. The more linear the output function of the image classifier is, the fewer iterations required for DeepFool to converge. Interestingly, despite that the adversarially-trained network has perturbations with larger $\ell_2$ norm, due to the more linear behavior of its output function, DeepFool converges faster on this network. In this sense, one can conclude that it is easier to attack adversarially-trained networks even in the white-box setting.

|  | **Med Iters** | **Max Iter** | $\ell_2$ **norm** |
|---|---|---|---|
| Regular He et al. (2016) | 4 | 15 | 0.209 |
| Adv. trained Madry et al. (2018) | 2 | 4 | 3.618 |

Table 2: DeepFool Moosavi-Dezfooli et al. (2016) performance on robust and regular ResNet-50 networks.

**DeepFool iterations behaviour** In this experiment, the goal is to qualitatively study the behaviour of output function along the trajectory of the iterations of DF for a single data point. In this case,

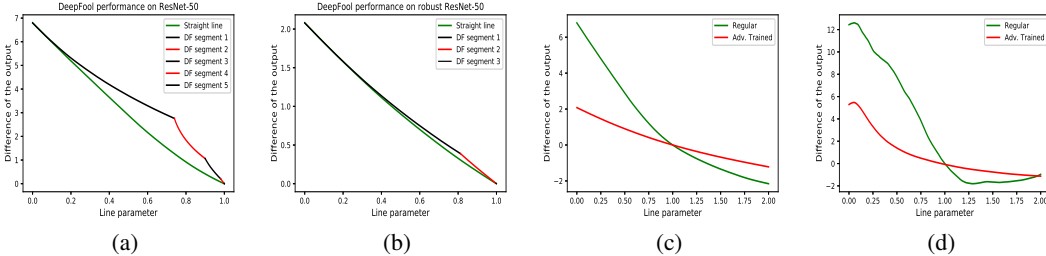

(a)      (b)      (c)      (d)

Figure 4: Performance evaluation of DeepFool over different iterations on (a) Regular ResNet 50 network, (b) Adversarially-trained ResNet 50 network. Differences of classifier's output for clean and adversarial labels when adversarial sample obtained by (c) DeepFool (close to the boundary), (d) Randomly using Gaussian perturbation (far from the boundary).

DeepFool requires 3 and 5 iterations to converge for robust and regular ResNet-50, respectively. We consider the difference of the logits corresponding to the original and the adversarial labels for our evaluation. We track this difference along two paths: 1) the straight path between the original image and the DF adversarial example (i.e., green line in Figs. 4a and 4b), and 2) the path taken by DF in each iteration (i.e. black and red line segments). We generate images on the line from the original image to the minimal perturbation adversarial example obtained by DeepFool. By varying the line parameter $t$, we consider the images along the line $\boldsymbol{x} = \boldsymbol{x}_0 + t(\boldsymbol{x}_{adv} - \boldsymbol{x}_0)$, where $t = 0$ corresponds to the original image and $t = 1$ gives the adversarial image which falls on the boundary. When the image is on the clean label side, the output value of the clean label is larger than the adversarial label. Approaching the boundary, this difference decreases where on the boundary the difference is equal to zero and the transition occurs. Assuming $\boldsymbol{x}_i$ as the output of DF in iteration $i$, each line segment $i$ (i.e. black and red segments in Figs. 4a and 4b) is corresponding to the images on the line $\boldsymbol{x} = \boldsymbol{x}_{(i-1)} + t(\boldsymbol{x}_i - \boldsymbol{x}_{(i-1)})$ for $t \in [0, 1]$, where $\boldsymbol{x}_i = \boldsymbol{x}_{adv}$ if $i$ is the last iteration. First, it can be seen that the straight path (green line) is much closer to the path constructed with DF iterations' segments for the adversarially-trained network compared to that of regular network. Second, it is shown that even in each line segment corresponding to each iteration traversed by DF algorithm, there is more non-linearity in regular networks as they are curved. As a result, although the adversarial training improved the robustness (increases the minimal $\ell_2$ norm), it provide an opportunity for the attacker to attack easier (with less number of iterations to converge) due to more linear behaviour of adversarially-trained networks.

**Non-linearity of the output** The goal is to see how the output of the classifier behaves when we push the image towards the boundary. In Fig. 4c the adversarial example is obtained using DeepFool, while in Fig. 4d the adversarial point is obtained with adding Gaussian noise along with binary search. In general, the difference of output in the adversarially trained network is more smooth and linear. This inherently shows that the no-linearity of the regular network is much higher than robust networks. Moreover, it can be seen that if the adversarial image is chosen randomly which is far from the original image (e.g. in Fig. 4d), this non-linearity is more sever. Thus, we can see that in the case of black-box attack, the attacker faces more non-linearity compared to the case of white-box setting. Since most of the black-box attacks try to obtain the normal vector to the boundary or estimate the gradient around a random boundary point, the more linear behaviour of output function and lower curvature of the boundary can help the adversary to better estimate the normal vector.

## 6 CONCLUSION

We showed that although the adversarial training is quiet effective against white-box attacks, in query-limited decision-based black-box attacks, it may not perform as efficiently as in the case for the white-box attacks. We demonstrated that since the adversarial training leads to a significantly flatter boundary and a more linear behavior of the image classifier, it can give an edge to certain types of black-box attackers whose goal is to estimate the normal vector to the boundary. This feature of the adversarially-trained networks can also provide a chance for minimal norm perturbation whit-box attacks to produce adversarial examples with less number of iterations. We introduced a highly parallelizable attack which can be successful against a similarity detector that can fool the robust network even easier compared to the regular classifier.

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

# A  APPENDIX

## A.1  MEAN DISTANCE OF THE QUERIES

In this section, our goal is to have an estimate for the optimal number of boundary points. The number of boundary points $M$ should not be too small resulting in close queries that will get detected by the similarity detector, and must not be too large that wastes the number of queries (as obtaining each boundary point costs $b$ queries on average).

We assume that the attacker has the information about $k$ and $\delta$. Thus, by computing the KNN distance in the design step of the attack, the attacker can evade the similarity detector. As mentioned previously, for a new query $i$, the $k$ nearest neighbour queries and their distances to $\boldsymbol{x}_i$ can be categorized into three types. In this section, we compute the average distance of each type of queries to $\boldsymbol{x}_i$.

- **Type I:** These queries are the ones required to obtain the boundary points. The boundary points can be obtained by starting from an adversarial perturbation with large $\ell_2$ norm and pushing it towards the boundary with binary search. The average distance of such queries with given query $\boldsymbol{x}_i$ can be computed as $\mu_1 = \frac{1}{b}\sum_{t=1}^{b} \|\boldsymbol{x}_i - \boldsymbol{x}_t\|$. Obtaining the exact $\mu_1$ can be complicated and not even necessary. Instead, we approximate it with $\mu_1 = \|\boldsymbol{x}_i - \boldsymbol{x}_{B,j}\|$ where $\boldsymbol{x}_{B,j}$ is the nearest boundary point to $\boldsymbol{x}_i$. Please note that $b$ is number of queries required for binary search. It could be determined after obtaining the first boundary point.

- **Type II:** The type II queries are the ones generated on the obtained boundary points to estimate the normal vector to the boundary. These queries are the most valuable queries in terms of information one can get from the image classifier. However, since their pairwise distances are small, they are the ones easily detected if not employed carefully. The means that we provide here are for a given type II query $\boldsymbol{x}_i \sim \mathcal{N}(\boldsymbol{x}_{B,j}, \sigma^2 \mathbf{I})$. The nearest queries of the type II $\boldsymbol{x}_t$ has the same multivariate Gaussian distribution $y = \boldsymbol{z}^T \boldsymbol{z}$ in which $\boldsymbol{z} =$

$\boldsymbol{x}_i - \boldsymbol{x}_t$ follows a Gamma distribution $Y \sim \Gamma(\alpha, \beta)$ with $\alpha = N/2$, $\beta = 4\sigma^2$. Thus, $d_{i,t} = \sqrt{y}$ has a $G \sim \text{Nakagami}(m, \Omega)$ distribution where $m = N/2$ and $\Omega = 2N\sigma^2$. Therefor the mean distance of the Type II queries is given by:

$$\mu_2 = \frac{\Gamma\left(m + \frac{1}{2}\right)}{\Gamma(m)} \left(\frac{\Omega}{m}\right)^{1/2}. \tag{7}$$

- **Type III:** Finally, type III includes the distance of queries between each group of queries on boundary points. These queries have large distance with each other which can increase the mean of the KNN similarity detector. We approximate it with the mean distance of the boundary points from one another. Thus, $\mu_3 = \frac{1}{M} \sum_{t=1}^{M} \|\boldsymbol{x}_{B,j} - \boldsymbol{x}_{B,t}\|$, $1 \leq t \leq M$.

Having the above information in hand, the attacker can have an estimate on the number of type II queries at each boundary point to get an estimation of the normal vector of the boundary without getting detected by the similarity detector. The number of queries at each boundary point should not be too large so that the similarity detector can simply detect the queries due to the small $d_i$. Moreover, it should not be too small to incur excessive overhead in finding boundary points. Thus, we have:

$$d_i = \frac{1}{k} \sum_{t=1}^{k} d_{i,t} = \frac{1}{k}(b\mu_1 + n\mu_2 + \gamma\mu_3) = \delta + \lambda, \tag{8}$$

where $\lambda$ is a margin to the threshold to make sure that the average distance can not be detected by the similarity detector. We may ignore the $n\mu_2$ term as it is typically small compared to $\mu_1$ and $\mu_3$. Having this approximation, the optimal value for $\gamma$ can be obtained as:

$$\gamma^* = \frac{k(\delta + \lambda) - b\mu_1}{\mu_3}. \tag{9}$$

It simply can be seen that if $\mu_3$ is large, then the number of type II queries $n$ per boundary point increases which results in more efficient deployment of queries. The optimal number of boundary points can be obtained by plugging $\gamma^*$ into equation 4 as $M^* = \frac{N-\beta}{k-b-\gamma^*}$. Based on this, by increasing the number of type III distances which results in increasing the KNN mean, one can see that the number of required boundary points to satisfy $d_i = \delta + \lambda$ decreases as well.

