# OpenReview forum: "On the exploitative behavior of adversarial training against adversarial attacks"
_ICLR.cc/2022/Conference — ICLR 2022 Submitted_

### Official Review · Reviewer_jHBK · 2021-11-01

**Correctness:** 2
**Technical Novelty And Significance:** 3
**Empirical Novelty And Significance:** 3
**Recommendation:** 5
**Confidence:** 4

**Main Review:**

## Strengths:
This paper makes bold claims: that it has developed a novel attack algorithm capable of defeating a recent defense, that this algorithm is more efficient and parallelizable, and that the robustness properties of adversarially trained networks diverge significantly from those of regularly trained ones. Each of these can be a strong contribution to the field of adversarial machine learning.

First, adaptive attacks are crucial to pushing our understanding of what “secure” machine learning is. By developing a method to defeat the kNN defense, this paper expands the understanding of what adversaries to defend against. Therefore, future work on defenses can use this one as a basis to build even more robust models . Second, the algorithm developed could be more efficient at generating black box adversarial examples. Black box attacks have so far had relatively limited real-world impact, as most machine learning systems do implement rate limiting. The idea of parallelizing the queries is an important contribution that expands the scope of methods for black box attacks. While the number of queries is still too large, this line of thinking is worth pursuing further and this paper offers a useful basis that the community would benefit from exploring further. Finally, the paper claims to have observed a paradoxical behavior in adversarilly trained networks. The “common wisdom” on those models is that they achieve better robustness at the cost of regular accuracy. Therefore, the observation that robustness of these models also suffers in the face of a new class of adversaries is an important one that the community should take note of.

## Weaknesses:
Unfortunately, this paper is lacking in the experimentation to back its claims. Most conspicuously, any mention of the success rate of the attacks developed is omitted. Instead, the authors use the magnitude of the perturbation as the basis for all claims given. While this can be an important secondary metric of how well an attacker is achieving their goals, it alone cannot back the claims made.  Concretely, this paper should be improved by performing all evaluations with a measure of how many adversarial examples successfully defeat the classifiers targeted.

It is possible that the authors implicitly assume that a reader should understand that all of their attacks have a 100% success rate. If that is the case, the authors should at a minimum explicitly state this. Furthermore, they should clarify after what number of queries this attack success rate is achieved. In an ideal evaluation, Figures 2 and 3 should be replotted with that measure.

Assuming these measurements back their claims, the authors could also be expanded in the following fashion:
* The claim for the linearity of the decision boundary of robust classifiers and  the implications for robustness against this parallelizable attack needs stronger backing. The authors should carry out the evaluation on multiple architectures, training and test sets, and tasks. It is insufficient to make such a strong claim based on a single network and a sample of the “standard” test set.
* The authors could do well to think more broadly about adversaries and robustness. While they include different attacks in their consideration (GeoDA and white box attacks), comparison against more attack methods could significantly strengthen the paper’s conclusions. These include adversaries that use different Lp norms in their constraints, more query-based black box methods, and non-gradient based adversaries (such as blurs, filters, overlays, and others).
* The authors could consider stronger defenses, including different settings of defenses they already consider. Would their attacks be defeated if the kNN defense method is tuned to a parallelizable attack? How does the success rate vary with the different parameters? Could defenders come up with entirely novel defenses in response to parallel attacks like the one proposed here?

**Summary Of The Paper:**

This paper proposes a new black box adversarial examples generation algorithm and uses it to draw conclusions about the decision boundary of adversarially trained models.

The algorithm developed is in response to a recent defense mechanism that stops black box attacks by detecting queries that are similar to each other. Therefore, this work proposes a new method that keeps similar queries to a minimum. The crux of the algorithm’s operation is that it estimates the normal vector to the decision boundary of the model – similar to prior work. Unlike previous papers, however, this one does so by querying at different spots along the decision boundary rather than focusing all the queries on the same point. This helps defeat similarity-based defenses.

The paper then applies this algorithm to both regularly trained networks and adversarially trained ones. This serves as the basis for claims about the geometry of the decision boundaries of each of those network types. To make these claims, it uses two metrics. First, this work observes the magnitude of the perturbation added to starting points in the adversarial example generation process against the number of queries the black box algorithm has used. Second, it introduces a new metric – robustness gain – that computes the ratio of that magnitude for adversarially robust networks and regular networks.

The conclusions of the work are all based on the difference of those metrics between adversarially trained and regular networks. Most interestingly, this paper claims that adversarial training is a weak defense against black box attacks of the type it develops.




**Summary Of The Review:**

The paper makes strong claims with important and interesting implications for the field of machine learning and develops a novel attack mechanism defeating a proposed defense. However, the claims are not backed up properly with a major, obvious metric missing entirely from the evaluation. If the claims are backed up according to the success rate of the attacks – instead of the magnitude of the perturbation – and if evaluation is carried out on a more diverse set of models and datasets, this can be a very strong paper.

---

### Official Review · Reviewer_knUq · 2021-11-03

**Correctness:** 3
**Technical Novelty And Significance:** 3
**Empirical Novelty And Significance:** 3
**Recommendation:** 5
**Confidence:** 4

**Main Review:**

[Strength]
1.	The paper provides a clever attack method that avoids detection based on the similarity among queries from a black-box attack.
2.	The formality of the problem definition and the approach looks to be good enough.

[Weakness]
1.	One of the experimental results can be interpreted differently so that one of the contributions can be just a misreading of the experimental results.
2.	Some parts of the paper require more explanations and formal descriptions about the concepts.

[Comments]
1.	About Figure 2b, as far as I understand, the authors tested two attack methods (single-point attack and multi-point attack) against two ResNet50 models (regular and robust) over the same set of target data points. (If they are tested on different sets of target data points, it could be an unfair comparison.) Then, against the regular ResNet50 model, the single-point attack can find much better perturbations (than the multi-point attack) for the same set of target data points. Doesn’t this just mean that “the multi-point attack is not effective against the regular ResNet50 model” rather than “adversarially trained model is more vulnerable (than the regular ResNet50 model) against the multi-point attack”? In other words, the main contribution that “certain black-box attack can perform better against adversarially trained model” is just a misreading of the experimental result and there is another possible interpretation that “the multi-point attack is not effective against the regular model”. (The reason for this another interpretation could be the reason that is provided by the authors; a regular model has a less smooth boundary.)
Also, the comparison with DeepFool (Figure 3b) says that the robustness gain is bigger for white-box attacks (constantly 17) compared to other black-box attacks. However, since the robustness gain is defined as a ratio, this value can be small just because black-box attacks cannot find a small enough perturbation for regular models. Again, we still don’t know whether this is because of the vulnerability of adversarially trained models or because of the poor performance of the black-box attacks.
I suggest the authors check the average perturbation sizes to show that the black-box attacks do not perform poorly against the adversarially trained model.
2.	First, robustness gain is only briefly described in the introduction and I cannot see any formal definition of it. Even though experiments are the only way to compute the robustness gain, it is better to describe it formally (an ideal & theoretical definition. If there is any reference that supports the concept, cite it.) and explain the intuition behind the concept.
Second, more details are needed for the robustness gain experiment. The main motivation for this experiment is unclear. $\eta$ (which must be the robustness gain) was not defined before it is mentioned.
3.	Lastly, I don’t think that robustness gain is a good measure to make comparisons. This is because the robustness gain is a ratio so it can be decreased by the poor performances of black-box attacks against a regular model. In other words, it is unclear whether the results come from the vulnerability of an adversarially trained model against black-box attacks or the results just come from the poor performance of the attacks against a regular model.


**Summary Of The Paper:**

First, this paper a new black-box attack method that can avoid a query similarity detector for an attempt to generate an adversarial example. Second, based on a set of experiments, the paper tries to show that an adversarially trained classifier is potentially more vulnerable against (black-box) adversarial attacks, which can be justified by the smooth decision boundary induced by the adversarial training.

**Summary Of The Review:**

While the new attack method seems interesting, it could be a hasty conclusion that the adversarially trained model is “more vulnerable” to black-box attacks. I suggest either making a weaker conclusion or strengthening the theoretical & experimental supports to the conclusion. Because of this concern, I’m a bit cautious to say that we should accept this paper.

---

### Official Review · Reviewer_ozXg · 2021-11-03

**Correctness:** 2
**Technical Novelty And Significance:** 2
**Empirical Novelty And Significance:** 2
**Recommendation:** 3
**Confidence:** 4

**Main Review:**

Overall, I find it difficult to understand the main focus of this paper. It seems to me that the novel black-box is the focus of this paper, but the title tells me it’s more about exploiting the smooth boundary of adversarially trained models.

There are a few concerns regarding this new attack. Firstly, the authors have made a rather unrealistic assumption that the attackers know the hyperparameters ($k$ and $\delta$) for the KNN detector. In practice, a black-box setting should only provide query access to attackers. In addition, the fact that this attack performs worse on regular DNN models is also a problem. In general, it seems this proposed attack is only useful when there is a KNN detector. An attacker will face a challenge if there is a KNN detector but the underlying model is a regular one, which could be a common scenario.

On the exploitation of adversarially trained models, I find the provided empirical evidence is not sufficient to draw the conclusion. The authors seem to take it as a fact that adversarially trained models have a flatter boundary. There is really no explanation in this paper on why it is true nor a reference that has shown this conclusion. Since almost all the arguments regarding the exploitation of adversarially trained models are based on this assumption, the authors should provide more explanation regarding it. Also, the fact that DeepFool requires 2 additional iterations on regular models is not a convincing evidence to show this boundary-based white-box attacks also exploit boundary smoothness.

**Summary Of The Paper:**

In this paper, the authors propose a new type of query-based black-box attack called multi-point attacks. This attack is designed to avoid being detected by KNN based detectors. Meanwhile, this paper also points that adversarially trained models have smooth boundary thus making it easier for attacks that could exploit this property to success.

**Summary Of The Review:**

I find a few issues in this paper that push me towards rejecting this paper. Please see comments above for details.

---

### Official Review · Reviewer_Q3xq · 2021-11-03

**Correctness:** 2
**Technical Novelty And Significance:** 2
**Empirical Novelty And Significance:** 2
**Recommendation:** 3
**Confidence:** 4

**Main Review:**

1. Novelty

The paper proposes a “multi-point” adversarial attack which works in black box hard label settings and could be parallelized. This is a novel attack, which to the best of my knowledge was not considered in literature before.

2. Assumptions about detectors of black box attack.

Authors study a setting when a defender tries to detect whether a query-based attack is done against their classifier and argue that their parallelizable attack could help to bypass such a detector. However authors impose several assumptions on such an attack which are not well justified in the paper.

First of all authors assume the black box setup for the classifier under attack, at the same time authors assume that adversary has the knowledge of parameters of the detector. Generally speaking, one would expect that these parameters would be unknown in black box setting. Thus further discussion is needed on how to estimate such parameters.

Additionally, the proposed attack samples several points near the decision boundary and then further explores regions around these points to estimate normal to the decision boundary. It seems like authors assume that these points near the decision boundary are far enough from each other, such that their exploration won’t trigger the detector. However, since all explored points are near original $x_0$ it might be the case they are all within $\delta$ range of the detector. In such a case multi-point attack does not really help preventing detection.

3. Evaluation of black box attack and adversarial robustness.

Authors claim that adversarially trained network might be easier to fool using black box attack. However their own experiments show that it’s not true in most cases (see Figure 3(a) and Table 1). There is a small part of figure 3 where this claim seems to be true (dashed blue line is above solid blue line), however it’s the weakest of the considered attacks.

Additionally, authors claim to evaluate their attacks on robust Resnet 50 and cite Madry et al paper. While Madry’s paper talks about adversarial training, it does not specifically talk about training of Resnet50 on Imagenet. Moreover it has been shown in future research by various people that adversarial training of Imagenet size models is not an easy task.
Thus authors should provide more details about training of their robust Resnet 50 model as well as it’s white box evaluation.

4. Overall the paper looks like a collection of several somewhat unrelated stories.

Specifically, part of the paper talks about multi-point attack and how to potentially evade detection. Another part of the paper talks about adversarial training and performance of black box attacks.
It might be beneficial to make a paper focused on one topic and study this topic in more detail and show stronger results.
Moreover, focusing paper on one topic would make it easier to read.


**Summary Of The Paper:**

The paper mainly talks about the following topics:
* Study whether adversarial training makes it easier to perform black box attack on the model.
* Novel multi-point black box attack, which is easy to parallelize and which is potentially can be more effective in the case when defender tries to detect black box attacks.


**Summary Of The Review:**

Paper proposed novel attack, however have several serious drawbacks:
* assumptions about attacker knowledge might be too strong and need further discussion
* evaluation of the black box attacks on adversarially trained model does not seem to support one of the claims of the paper.
* overall paper is written as a collection of somewhat unrelated stories.

---

### Official Review · Reviewer_S9ch · 2021-11-05

**Correctness:** 3
**Technical Novelty And Significance:** 2
**Empirical Novelty And Significance:** 2
**Recommendation:** 5
**Confidence:** 3

**Main Review:**

This paper considers a less studied aspect of adversarial attack that the input points are not independent and a defender can utilize this information for better detection. The proposed method is simple yet effective. However, I have the following concerns:
1. The paper only considers untargeted attacks. There are already many successful black-box untargeted adversarial attacks by utilizing the transferability between models. It is not very convincing why the query-based approach is better? On the other hand, targeted transferability is less significant, so I suggest that the authors also study whether the proposed approach is applicable to targeted attacks.
2. The authors only use a single-step attack, which can result in larger perturbation and thus may be detected. In this sense, although the proposed method can evade the detection from the similarity detector, it can be more easily detected by other detectors (e.g., anomaly detector).
3. I am skeptical whether a similarity detector is commonly used in real applications. This would require huge storage and computation cost, and it may cause some privacy issues for storing private user data.
4. Only one network architecture (ResNet-50) is used in experiments. Are the findings generalizable to other architectures?

**Summary Of The Paper:**

This paper proposes a query-based black-box adversarial attack that can only query a model with the class output.
The main consideration is to avoid sending very similar query points to evade the detection of a KNN-based detector.
The proposed method is parallelizable and less likely to be detected by a similarity-based detector.

**Summary Of The Review:**

Interesting idea but limited impact in real-world scenarios.

---

### Decision · Program_Chairs · 2022-01-20

**Decision:**

Reject

**Comment:**

The paper proposes a novel black-box attack aiming to fool a particular detector model. All reviewers see problems in the claims, the experiments etc and all argue for rejection. The authors did not provide a rebuttal to clarify any of the questions of the reviewers. Thus this is a clear reject.